# Nucleating Agents to Enhance Poly(l-Lactide) Fiber Crystallization during Industrial-Scale Melt Spinning

**DOI:** 10.3390/polym14071395

**Published:** 2022-03-29

**Authors:** Stefan Siebert, Johannes Berghaus, Gunnar Seide

**Affiliations:** Aachen-Maastricht Institute for Biobased Materials (AMIBM), Faculty of Science and Engineering, Maastricht University, Brightlands Chemelot Campus, Urmonderbaan 22, 6167RD Geleen, The Netherlands; stefan.siebert@maastrichtuniversity.nl (S.S.); j.berghaus@student.maastrichtuniversity.nl (J.B.)

**Keywords:** PLA melt spinning, crystallization, nucleating agents, BHET

## Abstract

The nucleating agent *N*,*N*′-bis(2-hydroxyethyl)-terephthalamide (BHET) has promising effects on poly(l-lactide) (PLA) under quiescent conditions and for injection molding applications, but its suitability for industrial-scale fiber melt spinning is unclear. We therefore determined the effects of 1% and 2% (*w*/*w*) BHET on the crystallinity, tenacity, and elongation at break of PLA fibers compared to pure PLA and PLA plus talc as a reference nucleating agent. Fibers were spun at take-up velocities of 800, 1400 and 2000 m/min and at drawing at ratios of 1.1–4.0, reaching a final winding speed of 3600 m/min. The fibers were analyzed by differential scanning calorimetry, wide-angle X-ray diffraction, gel permeation chromatography and tensile testing. Statistical analysis of variance was used to determine the combined effects of the spin-line parameters on the material properties. We found that the fiber draw ratio and take-up velocity were the most important factors affecting tenacity and elongation, but the addition of BHET reduced the mechanical performance of the fibers. The self-organizing properties of BHET were not expressed due to the rapid quenching of the fibers, leading to the formation of α′-crystals. Understanding the behavior of BHET in the PLA matrix provides information on the performance of nucleation agents during high-speed processing that will allow processing improvements in the future.

## 1. Introduction

Poly(l-lactide) (PLA) is an aliphatic thermoplastic polyester produced from renewable resources, with a production volume of ~400,000 tons in 2020 [1]. Pure PLA is often brittle, but it shows good elastic recovery, UV resistance and favorable burning behavior [2,3,4,5]. PLA is suitable for industrial composting and is therefore widely used in food containers and other single-use plastics. It is also suitable for melt spinning, and is therefore envisaged as an environmentally beneficial replacement for polyester filaments in clothing, carpets and upholstery that are currently derived from poly(ethylene terephthalate) (PET) [6,7,8]. PLA is also more comfortable than traditional polyester clothing because the fibers are light and only minimally absorb sweat, thus preventing odors [9,10].

In the absence of crystallization, the relatively low glass-transition temperature (T_g_) of PLA (55–60 °C) confers low heat resistance, which limits the adoption of industrial-scale processes [11]. Crystallinity increases heat resistance for subsequent textile processing [12,13], but the crystallization rate (R_c_) of PLA is low [14,15] and decreases further in the presence of greater amounts of d-lactide [16]. Several minutes are required for the crystallization of PLA when it is processed in bulk [17]. Crystallization is driven by C=O dipole interactions in the PLA backbone balanced by progressively diminishing molecular motion during cooling [18]. However, oxygen atoms also make the backbone more flexible, ensuring a high entropic crystallization barrier. Furthermore, crystals consist of helical chains that must be orientated to achieve optimal packing [2,18].

The R_c_ can be modified by uniaxial drawing during melt spinning. Strain-induced crystallization (SIC) occurs at high draw ratios [18,19,20], so melt-spun fibers cool much faster than bulk material. Optimal processing temperatures for SIC fall within the range of 65–80 °C [21]. Different winding speeds and draw ratios also influence fiber properties. The use of nucleating agents during SIC is expected to increase the degree of crystallinity (X_c_) at low draw ratios, in part by inducing earlier crystallization with lower degrees of molecular orientation, but the effect is negligible at high draw ratios [22,23]. The temperature window for PLA crystallization can be modified by using plasticizing agents that facilitate molecular movement at lower temperatures [2] and nucleating agents such as talc that promote the formation of nuclei or act as nuclei themselves [24,25,26,27].

Although many nucleating agents have been tested with PLA [26,27,28,29], few have been used for melt spinning [30]. Most were found to be slightly less efficient than talc, including chemically modified thermoplastic starch, cellulose nano-whiskers, and lignin derivatives [12,31,32]. TMC-328 not only acted as a nucleating agent but also triggered PLA degradation during processing [30,33,34]. The biobased and biodegradable nucleating agent orotic acid improved the R_c_ dramatically at levels as low as 0.3% (*w*/*w*) [28].

The PET-derivative *N*,*N*′-bis(2-hydroxyethyl) terephthalamide (BHET) can be synthesized entirely from renewable materials. BHET acts as a plasticizer in the melt before self-organizing into needle-like structures above the crystallization temperature (T_c_) of PLA. BHET has a high affinity for PLA and is therefore completely miscible in the PLA matrix, providing a large surface area for heterogeneous nucleation during bulk processing [35]. BHET has not yet been tested for high-velocity melt spinning, but we predicted that the replacement of talc with BHET would increase the X_c_ by altering the crystalline structure and therewith the crystallization behavior [36]. Accordingly, we introduced BHET as a nucleating agent and analyzed the resulting crystal structures in melt-spun fibers in order to better understand their behavior.

## 2. Materials and Methods

### 2.1. Materials

We used PLA grade Luminy L130 (Total-Corbion, Gorinchem, Netherlands), which is composed of <99% stereochemically pure poly(l-lactide) with a T_g_ of 55–60 °C and a melting point (T_m_) of ~175 °C. The nucleating agent BHET (Figure 1), with a T_m_ of 350 °C, was synthesized as previously described [35]. The melting behavior of BHET is unstable because it degrades directly after melting and must be dried before use. The talc we used as a reference nucleating agent was Plustalc H05 (Mondo Minerals, Amsterdam, The Netherlands) with a median particle size of 1.8 µm and a maximum particle diameter of 5.9 µm (as reported by the manufacturer).

Master batches of PLA containing 10% (*w*/*w*) talc or 5% (*w*/*w*) BHET were prepared using a Coperion twin-screw extruder equipped with gravimetrical dosing units. All materials were dried overnight at 60 °C in a vacuum before blending. Before spinning, the PLA granules were again dried overnight in a vacuum at 60 °C. The remaining moisture content was below 50 ppm, as determined by using a HydroTracer HT3 STD. Fiber samples were named according to the type and amount of nucleating agent (B = BHET, T = talc, PLA = none), the take-up velocity and the draw ratio (DR). For example, B1_800_DR2 indicates that the fiber contains 1% (*w*/*w*) BHET and was processed at a take-up velocity of 800 m/min with a final draw ratio of 2.

### 2.2. Fiber Spinning

PLA granules were filled into a hopper feeding the extruder under gentle nitrogen flow. When appropriate, the dried master batch was blended with PLA granules to achieve the specified ratio before filling the hopper.

Fibers were spun on an FET-100 Series multifunctional melt spinning system (Fiber Extrusion Technology, Leeds, UK) with an attached single-screw extruder set at 60 bar. The heating profile was set from 180 °C (hopper) to 220 °C (spinneret). Processing is usually carried out at 20–30 °C above the polymer T_m_ [37], but we have previously found that PLA processing is smoother with our machine at 220 °C (the upper end of the melt temperature range suggested by the manufacturer).

We set the material flow to 37.5 cm^3^/min with the spin pump running at 15 rpm. The spinneret contained 48 0.25-mm holes arranged in two circles. The filaments were quenched with air (23–24 °C) flowing at ~0.6 m/min. Spin-finish, containing fatty acid esters, surfactants, and antistatic agents (Zschimmer & Schwarz, Lahnstein, Germany) were applied to the fibers at 0.54 m^3^/min. Four heated godet duos were used to take up and draw the filaments (Figure 2).

Take-up velocities of 800, 1400 and 2000 m/min were applied at draw ratios of 1.1, 1.5, 2.0, 2.5, 3.0 and 4.0. Higher draw ratios were only attempted if fiber spinning was successful at the lower value. The draw ratio is the winding speed relative to the velocity of the take-up godet, and thus differs from the melt draw ratio (MDR), which is the exit velocity of the polymer melt from the spinneret relative to the velocity of the take-up godet. With our spinneret and process parameters, the take-up velocities of 800, 1400 and 2000 m/min yielded MDRs of 50, 88 and 126, respectively.

The set take-up velocity determined the velocity of godet duo 1, so the other godet duos were freely adjusted to create and maintain a stable process, with most of the fiber drawing occurring between godet duos 2 and 3. The default temperatures were set to 50, 80, 80 and 70 °C for godet duos 1–4, respectively. Some process parameters were adjusted slightly to accommodate changes in the nucleating agent content, and the take-up velocities were modified to ensure smooth-running process conditions.

Fiber tension was monitored using a digital tension meter and was adjusted to remain at ~10 cN during winding. The fiber was wound onto bobbins using the industrial winder WinTens 602 (STC Spinnzwirn, Chemnitz, Germany) factory-optimized for winding velocities of 500–4200 m/min. The bobbins were wound for 5–7 min to ensure process stability.

### 2.3. Tensile Testing

Tensile testing was carried out using a ZwickiLine Z2.5 (Zwick Roell, Ulm, Germany) fitted with an Xforce HP load cell for a 50 N nominal force prepared with capstan clamps for fiber testing. Fibers were tested with a starting length of 200 mm and a pretension of 0.1 cN/tex. Testing was performed at a draw rate of 200 mm/min according to DIN EN ISO 5079. Before each test, the linear density of the spun fibers (dtex) was determined gravimetrically as the average of five 100 m samples according to the DIN EN ISO 1973 standard. The linear density was tested in triplicate.

### 2.4. Gel Permeation Chromatography

The molecular weight of the fibers was determined by gel permeation chromatography (GPC) using a 1260 Infinity II SECurity device (Agilent Technologies, Santa Clara, CA, USA) with a coupled refractive index detector. The GPC instrument was calibrated using polymethyl methacrylate (PMMA) standards. Samples were prepared by dissolving ~4 mg of fiber in 1.5 mL hexafluoroisopropanol (HFIP).

### 2.5. Differential Scanning Calorimetry

For differential scanning calorimetry (DSC), 3.5 mg samples were placed in pierced aluminum pans on a DSC 214 device (Netzsch, Selb, Germany). We applied a single heating cycle of 30–230 °C at 10 K/min, followed by an isothermal hold at 230 °C for 3 min before cooling to 30 °C at 10 K/min. Samples of pure PLA and those containing 1% (*w*/*w*) of each nucleation agent prepared at a take-up velocity of 800 m/min and a draw ratio of 1.1 were tested in triplicate, whereas all other samples were tested in single experiments. The data were analyzed using Proteus Thermal Analysis v7.0.1 (Netzsch). The caloric data from the DSC measurements allowed us to calculate X_c_ values if the maximum attainable heat of fusion (ΔHm^0^) for a single crystal of pure poly(l-lactide) was also known. We therefore used the reported value of 93 J/g [2,18,38,39,40,41,42,43] and calculated X_c_ using Equation (1):X_c_ = (ΔH_m_ − ΔH_cc_)/ΔH_m_^0^(1)

To account for the quantity of nucleating agents incorporated into the polymer melt, the nominal X_c_ value was multiplied by the amount of PLA (0.99 for 1% BHET and talc, or 0.98 for 2% BHET and talc) to give the final X_c_ value.

### 2.6. Wide-Angle X-ray Diffraction

Samples were prepared from 3–4 loops of fiber to gain sufficient intensity for the wide-angle X-ray diffraction (WAXD) scan. The straightened fibers were analyzed at DSM (Geleen, The Netherlands) on a SAXSLAB Ganesha System (Saxslab, Kopenhagen, Denmark). Cu radiation (λ = 0.15406 nm) was used at a detector distance of ~0.08 m. The refractive intensities within angles of 2–27° were recorded at 0.0025° intervals. Fityk v1.3.1 was used for peak deconvolution [44] in the exported 2θ curves.

## 3. Results and Discussion

### 3.1. Preliminary Adjustments

The polymer throughput for all trials was a constant 37.5 cm^3^/min. At a draw ratio of 1.1 and the slowest take-up velocity of 800 m/min, the linear density of the fiber exceeded 500 dtex. At the fastest take-up velocity of 3600 m/min, the linear density was 127 dtex. When we added BHET, the polymer melt started dripping from the spinneret due to an increase in viscosity. The spin pump pressure decreased by 10 bar when the polymer contained 1% (*w*/*w*) BHET (compared to pure PLA or PLA + 1% (*w*/*w*) talc). At 2% (*w*/*w*) BHET, it was necessary to reduce the temperature by 20 °C in order to spin fibers successfully. We also needed to reduce the temperatures of godet duos 2 and 3. The correct spin-finish grade of PLA is necessary to ensure a stable process and smooth fibers.

### 3.2. Tensile Testing

The fibers were tested as-spun without additional tempering. The tensile strength of the fibers increased with faster take-up velocities and higher draw ratios (Figure 3). This refined the molecular orientation, making more tie molecules available to link the separate crystals [22,45]. The effect was clearest for the samples containing 1% (*w*/*w*) talc, although the tenacity of these fibers at take-up velocities of 800 and 1400 m/min was lower than that of pure PLA fibers.

Under comparable processing conditions with a take-up velocity of 800 m/min, pure PLA fibers showed the highest tenacity and the samples containing BHET showed the lowest tenacity (Figure 4). Increasing the take-up velocity and/or the draw ratio did not improve the tenacity of BHET-containing fibers beyond a value of 21 cN/tex. There was little difference in tenacity between samples containing 1% and 2% (*w*/*w*) talc, although the latter could be extended further if production was successful. Data point values and the corresponding error values are given in Table 1.

Having tested various parameters side by side, an analysis of variance (ANOVA) was used to evaluate the combinations of effects (Figure 5, Table 2). A custom plan was created with the following factors: *additive* (talc or BHET), *amount* (0%, 1% or 2%, where 0% represents pure PLA), *take-up velocity* (800, 1400 or 2000 m/min) and *draw ratio* (1.1, 1.5 or 2.0). Most combinations could be produced. Figure 5a shows a Pareto chart ranking the effective strength of each factor and the interactions that influenced tensile strength. The take-up velocity and draw ratio were found to be the most significant variables (92% of the cumulative percentage), indicating that molecular orientation has the strongest impact on tenacity. Furthermore, the amount and presence of the nucleation agent were significant, as were the interactions *additive*amount*, *take-up velocity*draw ratio*, *amount*take-up velocity* and *amount*additive*draw ratio*. Figure 5b shows a Pareto chart ranking the effective strength of each factor and the interactions that influenced elongation and indicated that the draw ratio was more significant than the take-up velocity in this case, and that no other factors played a significant role.

### 3.3. Gas Permeation Chromatography

To investigate the degradation of the compounds during processing, GPC analysis was applied only to those fiber samples containing the highest amount of additive and that experienced the highest strain during spinning due to the take-up velocity and draw ratio. GPC analysis revealed a small reduction in the molecular weight of the processed samples compared to unprocessed PLA (Figure 6, Table 3). This was anticipated and reflects the elevated temperature and shear stress during compounding and the melt spinning process. The molecular weight of the processed PLA samples containing talc remained the same as the processed samples of pure PLA, indicating that talc does not trigger PLA degradation during processing. The molecular weight of the samples containing 2% (*w*/*w*) BHET was slightly lower than that of the other processed samples, in agreement with a previous study using this additive [35]. However, this effect was more prominent in our process, reflecting a longer exposure of the samples to high temperature and shear stress in our industrial-scale device compared to the laboratory equipment used for the earlier study. A small reduction in molecular weight during processing is not expected to significantly influence the mechanical performance of the fibers or their crystallization behavior.

### 3.4. Differential Scanning Calorimetry

The spun fibers containing BHET yielded DSC curves similar to those previously reported [35] for pure PLA (Figure 7, Table 4). Analysis of the cold-crystallization peak for 1% (*w*/*w*) BHET under different processing conditions revealed that crystallization takes longer for the blends with higher cold-crystallization temperatures (T_cc_) (Figure 7a). At a slow take-up velocity of 800 m/min and a low draw ratio, the T_cc_ peak was found at 93 °C. As the draw ratio was increased, the peak temperature fell to 76 °C, and if the take-up velocity was increased while maintaining a medium draw ratio, the peak temperature fell further to 74.6 °C. A lower temperature was sufficient to form crystals if the polymer chains were highly oriented. At a medium take-up velocity (1400 m/min) and low draw ratio, the T_cc_ was similar to the slow take-up sample and shows that a low T_cc_ is maintained as the draw ratio increases. The higher the draw ratio, the earlier crystallization begins, which suggests that stretching in the solid state leads to rapid SIC.

Fibers spun without a high draw ratio featured a small exothermic crystallization peak immediately before the endothermic melting peak (Figure 7b). This is associated with recrystallization where α′-crystals transform into α-crystals [43]. Spinning with low draw ratios leads to the formation of imperfect α′-crystal structures due to the low strain during drawing. These α′-crystals can melt and recrystallize to form α-crystals. The exothermic peak may also reflect a high level of polymer orientation in the amorphous phase due to the strain applied while drawing from the melt, which is thought to promote the formation of a mesomorphic phase [46,47].

The exothermic peak vanished at higher draw ratios, but a twin peak developed for the T_m_. This was due to the highly oriented polymer chains, resulting in the formation of imperfect crystals that melted at a slightly lower temperature than perfect crystals. This phenomenon was exacerbated by rapid quenching at high velocities, which provided insufficient time to form perfect crystals.

There were no significant differences between the heating cycles for pure PLA and PLA plus either of the additives at the slowest take-up velocity and lowest draw ratio. At higher take-up velocities, the T_c_ declined for pure PLA and PLA + talc but not for PLA + BHET, indicating that BHET cannot exploit the higher degree of polymer orientation in the fiber, thus preventing crystallization at lower temperatures (Figure 8, Table 5). This is the opposite of the behavior anticipated for a nucleating agent.

During the cooling cycle, the presence of talc increased the T_c_ by almost 10 °C. This effect has been reported before and is often used to accelerate crystallization [31,32].

The enthalpy of recrystallization (exothermic peak from the first heating cycle) was subtracted from the melting enthalpy to calculate X_c_, which generally increased at higher draw ratios due to the higher degree of polymer orientation and SIC during spinning (Figure 9). As stated above, X_c_ was corrected by the amount of additive in each composite material. At low take-up velocities, X_c_ declined as the draw ratio increased from 1.5 to 2, but it increased more rapidly at higher draw ratios. The only samples that did not show a decrease in X_c_ within this range of draw ratios were those containing 2% (*w*/*w*) talc, indicating that the talc particles suppress this effect if enough nuclei are present to maintain the crystallization process.

A decrease in X_c_ was even observed at medium take-up velocities, especially for pure PLA fibers. The PLA fibers with BHET also showed a decrease in X_c_ but at a slightly higher draw ratio, possibly reflecting the switch from drawing in the molten state (as defined by the take-up velocity) to solid-state drawing after consolidation. Alternatively, this is the point at which there is insufficient time for crystallization and the stress is not yet high enough for SIC. There was no decrease in X_c_ at higher take-up velocities, which strengthens the hypothesis that SIC occurs under these conditions. The maximum X_c_ was ~55% for the selected ΔHm^0^ and is clearly shown for the fibers containing 2% (*w*/*w*) talc (T2_800) in Figure 9a. The plateau is consistent with earlier reports [20].

The DSC data were statistically evaluated by ANOVA (Figure 10, Table 6). The main effects contributing to the X_c_ were ranked by importance, as follows: draw ratio, take-up velocity, type and amount of additive. This confirms that the draw ratio has the strongest effect on the X_c_.

### 3.5. Wide-Angle X-ray Diffraction

The WAXD data were deconvoluted to isolate the crystalline signals, which are shown in Figure 11. It is clear that α-crystals (small peaks at 16.4°) only formed at high draw ratios, whereas faster take-up velocities only increased the meso-phase [46]. This supports the importance of a high draw ratio during spinning for crystal formation. At high draw ratios, a second peak developed at 18.7° for PLA and PLA + talc, indicating the presence of α′-crystals. The addition of talc changed the crystallization behavior because the peak at 16.4° indicates that α-crystals were formed even at lower draw ratios. In contrast, the presence of BHET strengthened the meso-phase in comparison to pure PLA.

The deconvoluted WAXD signals for fibers prepared at a slow take-up velocity and high draw ratio revealed that PLA containing 1% (*w*/*w*) BHET had a slightly higher peak than pure PLA, but a lower peak than PLA containing 2% (*w*/*w*) talc (Figure 11d). Pure PLA and the fibers containing 1% (*w*/*w*) talc also showed a small peak at 18.7°, indicating the presence of α′-crystals. This peak disappeared from the fibers containing 2% (*w*/*w*) talc, probably reflecting smaller crystals originating from the more abundant nuclei. Because the process conditions were identical, a high level of strain in the fiber led to imperfect larger crystals but had a less significant effect on the small crystals, which prevented the formation of α′-crystals.

## 4. Conclusions

BHET worked as a plasticizer in the molten PLA as expected, but the presence of 2% (*w*/*w*) BHET reduced the melt viscosity and thus prevented effective melt spinning. However, BHET did not enhance the tensile strength of the spun fibers and even reduced their mechanical performance compared to pure PLA or PLA + talc. Statistical analysis revealed that the additives had no significant effect on the tenacity or elongation of the as-spun fibers, and GPC did not reveal any substantial polymer degradation during processing. Although BHET is known to promote hydrolysis, the minimal degradation we observed is not expected to influence the mechanical properties of the resulting fibers. BHET was not an efficient nucleating agent under our melt spinning conditions. The BHET-loaded fibers behaved like pure PLA fibers with only slightly lower values, and there was no change in the final X_c_ or T_c_. Importantly, with higher take-up velocities, BHET inhibited crystallization at lower temperatures. WAXD peaks representing α-crystals only appeared at high draw ratios for all three types of material. BHET may be a suitable nucleating agent at low take-up velocities and low draw ratios when the temperature is high, but this would result in a much lower production efficiency. We conclude that the BHET content should be tailored for each polymer to enable dissolution at higher temperatures in the polymer melt. This will facilitate subsequent compounding and melt spinning at lower temperatures without destroying the self-assembled structures of the nucleating agent.

## Figures and Tables

**Figure 1 polymers-14-01395-f001:**
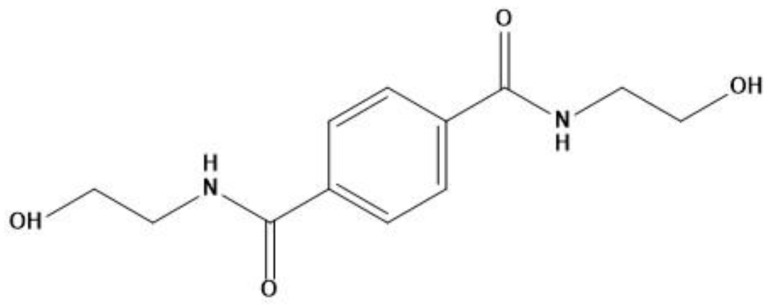
Chemical structure of *N*,*N*′-bis(2-hydroxyethyl) terephthalamide (BHET).

**Figure 2 polymers-14-01395-f002:**
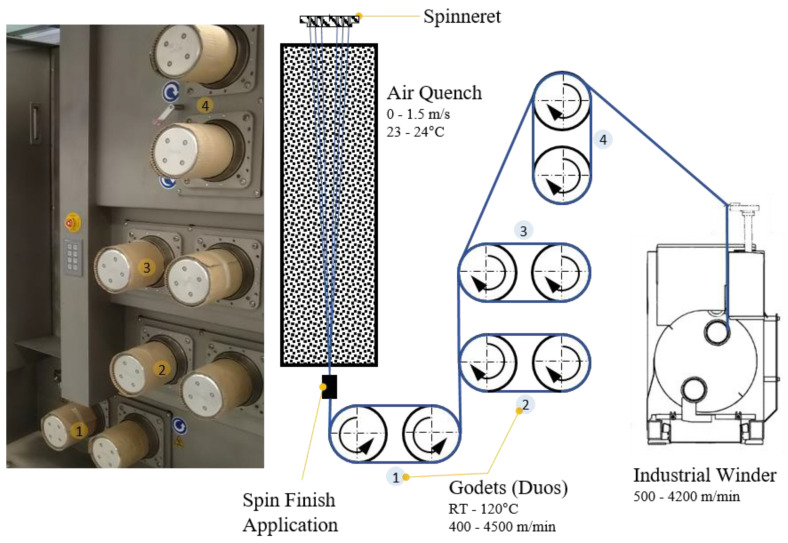
Industrial melt spinning plant FET-100 (**left**) and the corresponding schematic process (**right**).

**Figure 3 polymers-14-01395-f003:**
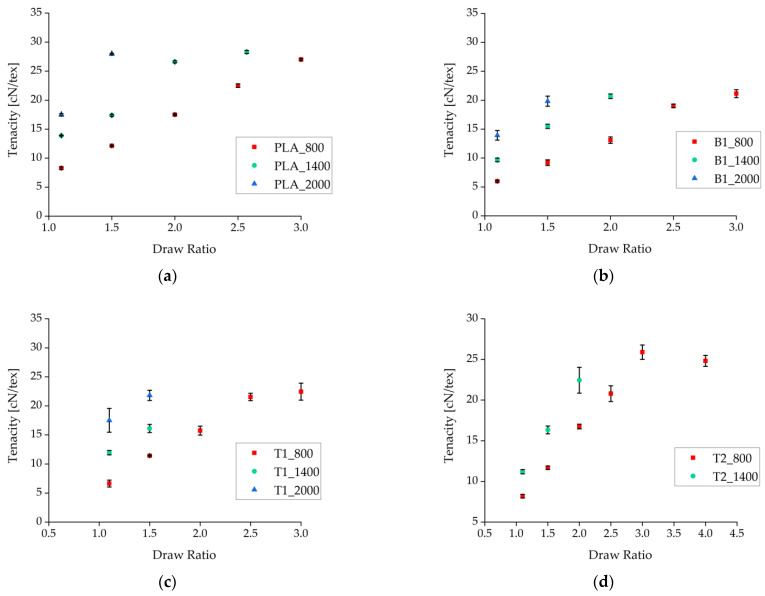
Tenacity of fibers plotted against the draw ratio for different nucleation agents. (**a**) Pure PLA fiber as a reference. (**b**) Fibers containing 1% (*w*/*w*) BHET. (**c**) Fibers containing 1% (*w*/*w*) talc. (**d**) Fibers containing 2% (*w*/*w*) talc. Data are plotted as the mean ± standard deviation (n = 5).

**Figure 4 polymers-14-01395-f004:**
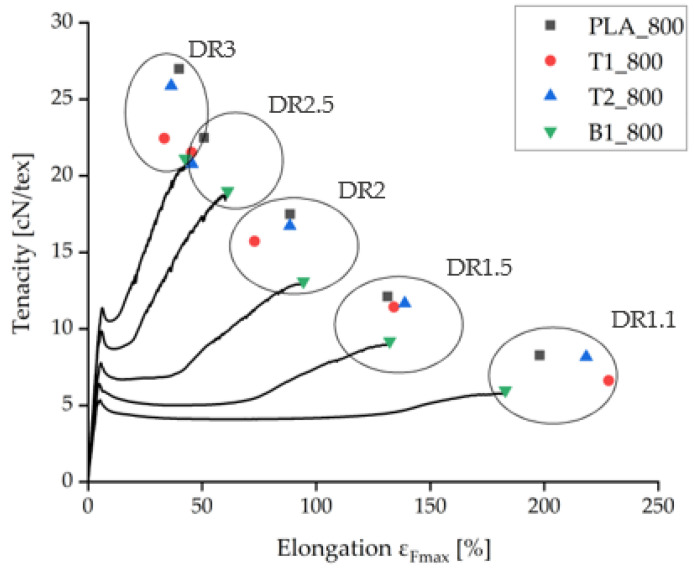
Tensilegram for as-spun fibers at 800 m/min take-up velocity and various draw ratios. The complete measurement curves are shown for samples containing 1% BHET (B1_800).

**Figure 5 polymers-14-01395-f005:**
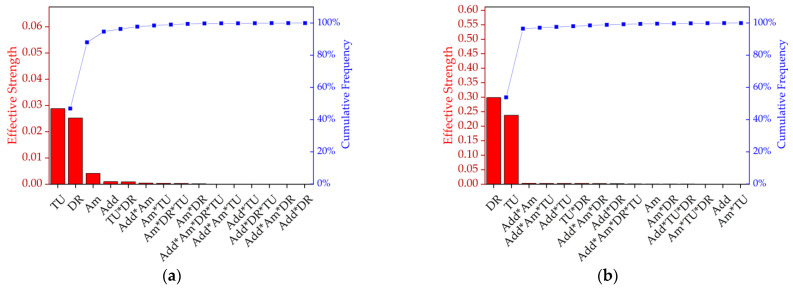
Pareto chart ranking the effective strength and cumulative frequency of the factors and the interactions that influence (**a**) tenacity and (**b**) elongation. DR = draw ratio, TU = take-up velocity, Add = nucleating agent additive, and Am = amount of nucleating agent.

**Figure 6 polymers-14-01395-f006:**
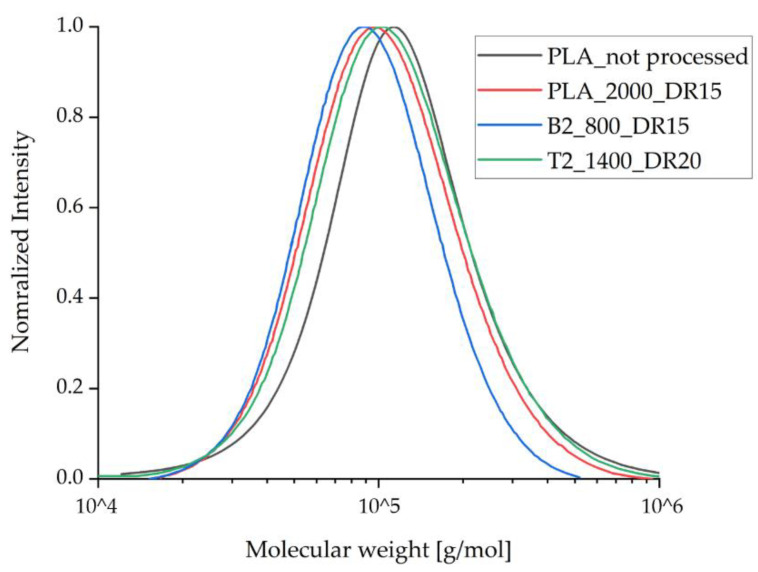
Molecular weight distribution for the processed fibers compared to pure PLA, as determined by gel permeation chromatography. The values for Mw and Mn are listed in Table 3.

**Figure 7 polymers-14-01395-f007:**
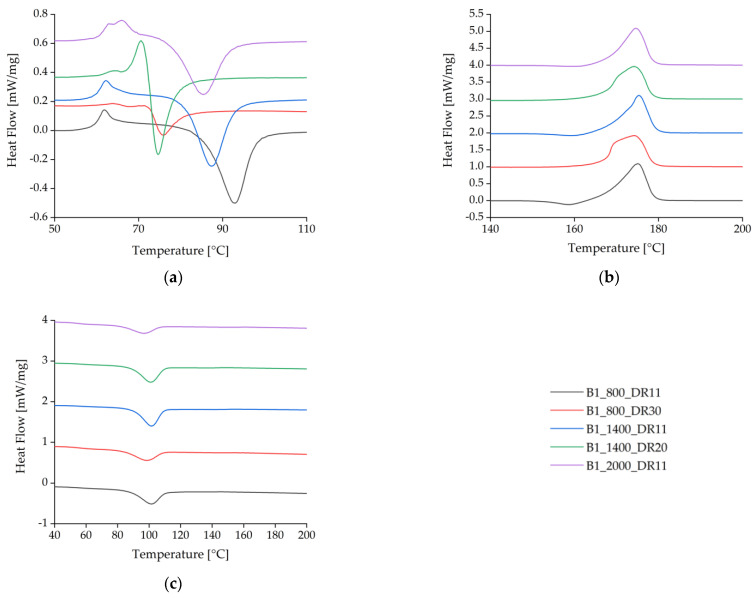
Differential scanning calorimetry thermograms for the first heating cycle of BHET-containing fibers at different take-up velocities and draw ratios. (**a**) Detailed view of the glass-transition temperature T_g_ and cold-crystallization temperature T_cc_. (**b**) Melting temperature T_m_. (**c**) Crystallization temperature T_c_ during cooling.

**Figure 8 polymers-14-01395-f008:**
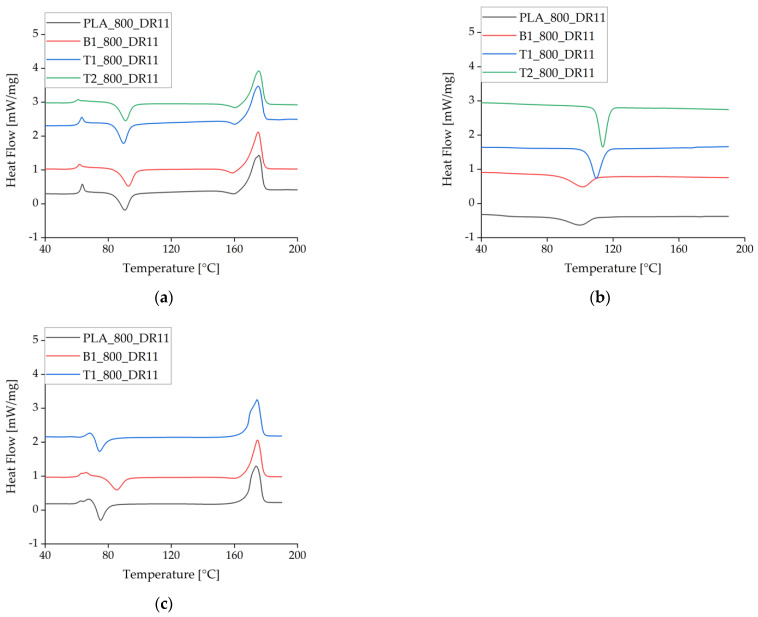
Differential scanning calorimetry data (exotherm down) for the first heating cycle at different take-up velocities and in the presence of different nucleating agents. (**a**) The first heating cycle at a take-up velocity of 800 m/min. (**b**) The corresponding crystallization peak during cooling. (**c**) The heating cycle at a take-up velocity of 2000 m/min. The curves are offset for visual clarity.

**Figure 9 polymers-14-01395-f009:**
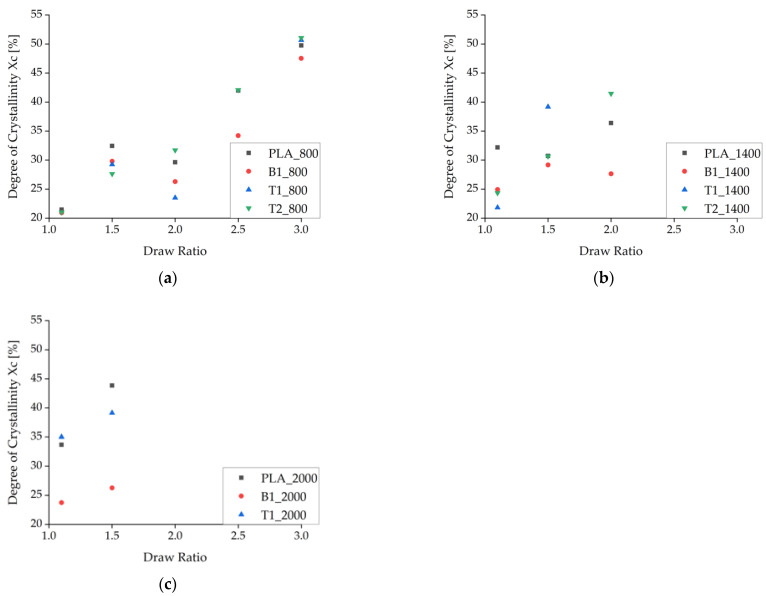
Degree of crystallinity (X_c_) calculated from enthalpy as the draw ratio increases. The data represent take-up velocities of (**a**) 800 m/min, (**b**), 1400 m/min, and (**c**) 2000 m/min.

**Figure 10 polymers-14-01395-f010:**
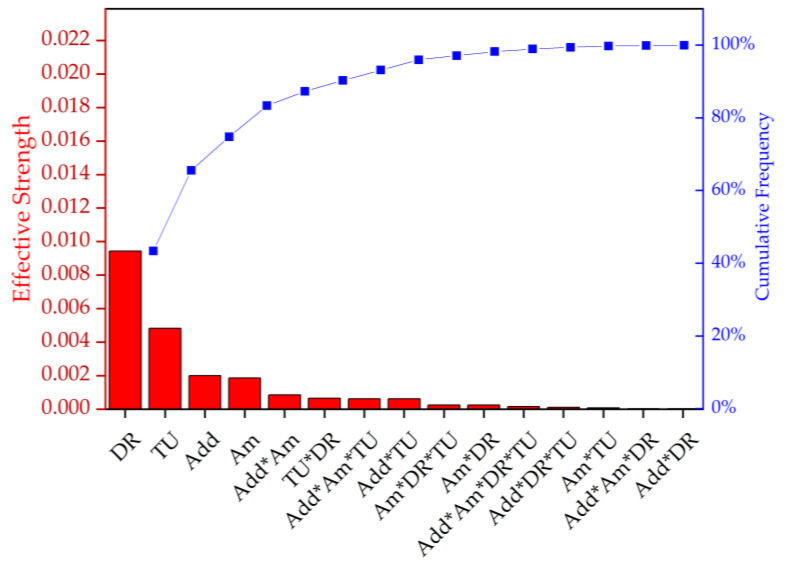
Pareto chart representing the effective strength and cumulative frequency of the factors and their interactions on the degree of crystallinity (X_c_). DR = draw ratio, TU = take-up velocity, Add = nucleating agent, and Am = amount of nucleating agent.

**Figure 11 polymers-14-01395-f011:**
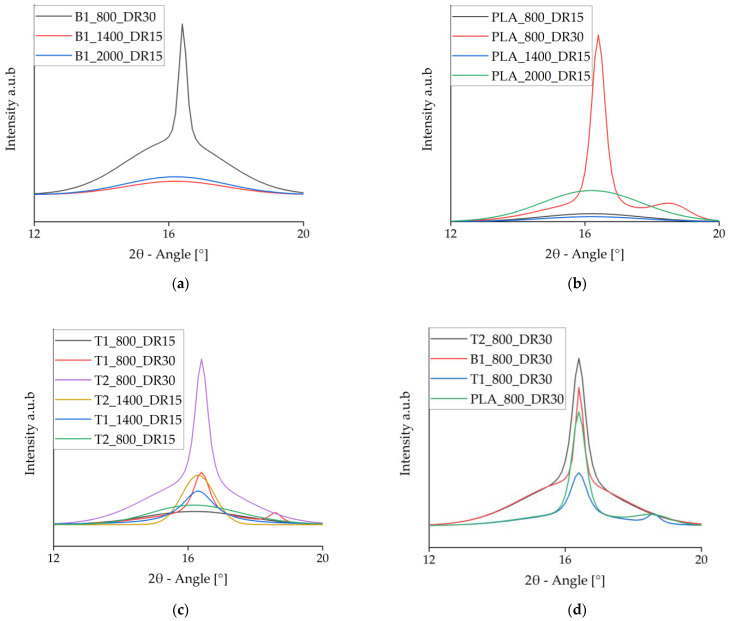
Intensity vs. 2θ curves of the crystalline phase at 2θ = 16.4° from deconvoluted wide-angle X-ray diffraction data. (**a**) Samples containing BHET. (**b**) Pure PLA samples. (**c**) Samples containing talc. (**d**) Comparison of pure PLA and samples containing each nucleating agent.

**Table 1 polymers-14-01395-t001:** Data values for the samples displayed in Figure 4.

Sample	Tenacity		Elongation	Sample	Tenacity F_max_		Elongation
	F_max_ [cN/tex]	SD	ε_Fmax_ [%]		F_max_ [cN/tex]	SD	ε_Fmax_ [%]
PLA_800_DR11	8.3	0.171	198.1	B1_800_DR11	6.0	0.121	183.0
PLA_800_DR15	12.1	0.103	131.3	B1_800_DR15	9.2	0.486	132.3
PLA_800_DR20	17.5	0.110	88.6	B1_800_DR20	13.1	0.556	94.4
PLA_800_DR25	22.5	0.298	50.9	B1_800_DR25	19.0	0.312	61.2
PLA_800_DR30	27.0	0.142	39.9	B1_800_DR30	21.1	0.684	42.6
T1_800_DR11	6.6	0.598	228.3	T2_800_DR11	8.2	0.228	218.4
T1_800_DR15	11.4	0.089	134.1	T2_800_DR15	11.7	0.214	138.9
T1_800_DR20	15.7	0.772	73.0	T2_800_DR20	16.7	0.288	88.5
T1_800_DR25	21.5	0.634	45.4	T2_800_DR25	20.8	0.969	45.7
T1_800_DR30	22.5	1.460	33.4	T2_800_DR30	25.9	0.888	36.5

**Table 2 polymers-14-01395-t002:** Significance level (p) of factors and their interactions corresponding to Figure 5.

Factor	Significance Level (p) Tenacity	Significance Level (p) Elongation
Add	0.000	0.033
Am	0.000	0.874
TU	0.000	0.000
DR	0.000	0.000
Add*Am	0.000	0.140
Add*TU	0.056	0.000
Add*DR	0.073	0.004
Am*TU	0.000	0.021
Am*DR	0.159	0.981
TU*DR	0.000	0.001
Add*Am*TU	0.122	0.000
Add*Am*DR	0.040	0.126
Am*DR*TU	0.303	0.091
Add*Am*DR*TU	0.375	0.877

**Table 3 polymers-14-01395-t003:** Mw, Mn and polydispersity values for the samples tested in Figure 6.

Sample	Mw [g/mol]	Mn [g/mol]	Polydispersity
PLA_Reference	1.51 × 10^5^	9.70 × 10^4^	1.56
PLA_2000_DR15	1.27 × 10^5^	8.71 × 10^4^	1.46
B2_800_DR15	1.10 × 10^5^	7.58 × 10^4^	1.45
T1_2000_DR15	1.32 × 10^5^	8.79 × 10^4^	1.50
T2_1400_DR20	1.37 × 10^5^	8.95 × 10^4^	1.53

**Table 4 polymers-14-01395-t004:** Differential scanning calorimetry data for Figure 7.

Sample	T_g_ [°C]	T_cc_ [°C]	T_m_ [°C]	T_c_ [°C]
B1_800_DR11	65.2	92.0	174.3	99.7
B1_800_DR30	66.0	76.0	174.3	98.4
B1_1400_DR11	60.9	87.3	175.4	101.3
B1_1400_DR20	62.3	74.6	174.3	100.9
B1_2000_DR11	63.6	85.4	174.6	96.6

**Table 5 polymers-14-01395-t005:** Differential scanning calorimetry data for Figure 8.

Sample	T_g_ [°C]	T_cc_ [°C]	T_m_ [°C]	T_c_ [°C]
B1_800_DR11	65.2	92.0	174.3	99.7
PLA_800_DR11	61.2	90.8	174.7	102.9
T1_800_DR11	65.0	90.0	174.7	110.0
T2_800_DR11	59.2	91.0	175.4	113.7
B1_2000_DR11	63.6	85.4	174.6	96.6
PLA_2000_DR11	61.1	75.1	173.8	102.0
T1_2000_DR11	59.4	74.5	174.4	112.2

**Table 6 polymers-14-01395-t006:** Significance level (p) of factors and their interactions corresponding to Figure 10.

Factor	Significance Level (p)
Add	0.000
Am	0.888
TU	0.000
DR	0.000
Add*Am	0.003
Add*TU	0.010
Add*DR	0.614
Am*TU	0.188
Am*DR	0.073
TU*DR	0.809
Add*Am*TU	0.214
Add*Am*DR	0.799
Am*DR*TU	0.199
Add*Am*DR*TU	0.831

## Data Availability

The datasets used and/or analyzed during this study are available from the corresponding author on reasonable request.

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
