# Peer review of "Nucleating Agents to Enhance Poly(l-Lactide) Fiber Crystallization during Industrial-Scale Melt Spinning"

_polymers, 2022, doi:10.3390/polym14071395_

Round 1

Reviewer 1 Report

1. In figure 3 and figure 9, the samples were tested at different draw ratios. For example, in figure 3a, PLA1400 did not have the data at the draw ratio of 3.0. In figure 3b, B1-1400 lack of the data at the draw ratio of 2.5 and 3.0. and etc. The missing data will readers difficult to compare the samples with each other.
2. The reply for comment 2 is wrong. In line 91, the content of BHET is 1%.
3. Error bar is missing in the Table 1.
4. The figure 5 has no meaning.
5. The reply for Comment 8 and Comment 9 is not acceptable.

Author Response

#

Comment

Rebuttal

1

1. In figure 3 and figure 9, the samples were tested at different draw ratios. For example, in figure 3a, PLA1400 did not have the data at the draw ratio of 3.0. In figure 3b, B1-1400 lack of the data at the draw ratio of 2.5 and 3.0. and etc. The missing data will readers difficult to compare the samples with each other.

As explained in the previous revision, some parameters were not compatible with fiber spinning. It is not possible to present the data for fibers that cannot be spun, hence their absence from the corresponding figures.

2

The reply for comment 2 is wrong. In line 91, the content of BHET is 1%.

We thank the reviewer for highlighting this error and we have revised the manuscript accordingly by correcting the calculation in the methods section, adding explanatory statements in the methods and results (lines 159-161), and adjusting the corresponding figure.

3

Error bar is missing in the Table 1.

We have added standard errors to the table as requested.

4

The figure 5 has no meaning.

Figure 5 clearly demonstrates that the incorporated nucleating agents do not have major effects on the tenacity and elongation of the spun fibers.

5

The reply for Comment 8 and Comment 9 is not acceptable.

We consider it unconstructive to provide such a statement as a fait accompli with no explanation as to why the previous responses were unacceptable, therefore giving us no fair opportunity to address the underlying issue.   

Reviewer 2 Report

The research paper idea is novel and the data presentation is very easy to follow up. The only thing I want to be changed is data representations. I recommend using professional drawing platforms like "OriginLab" instead of Excel.

Author Response

#

Comments Round 2

Rebuttal 2

1

The research paper idea is novel and the data presentation is very easy to follow up. The only thing I want to be changed is data representations. I recommend using professional drawing platforms like "OriginLab" instead of Excel.

We very much appreciate the reviewer’s comments and suggestions, and have redrawn all the figures in OriginLab as requested.

Reviewer 3 Report

The article is fundamentally interesting and describes the mechanical properties of polylactide fibers after the addition of bis(2-hydroxyethyl-terephthalamide) as a crystallization aid.

Unfortunately, the article cannot be published in this form.

  1. The Manuscript is much too long can be shortened by about 90% (use of supporting information) and thus reduced to essential statements.
  2. Please demonstrate how far the dihydroxy compound is incorporated into the polylactide chain. 
  3. Is there an influence of the additive on the molar mass: please discuss

Author Response

#

Comment

Rebuttal

1

The article is fundamentally interesting and describes the mechanical properties of polylactide fibers after the addition of bis(2-hydroxyethyl-terephthalamide) as a crystallization aid.

Unfortunately, the article cannot be published in this form.  

2

    The Manuscript is much too long can be shortened by about 90% (use of supporting information) and thus reduced to essential statements.

It is unrealistic to shorten any manuscript by 90%, regardless of how much material can be moved to supplements. The reviewer seems to be asking for a summary report of essential statements which is not the format of long-form articles in Polymers.

3

Please demonstrate how far the dihydroxy compound is incorporated into the polylactide chain.

This appears to be a misunderstanding on the part of the reviewer. BHET molecules are not incorporated into the PLA chain, but instead are distributed between the chains in the melt. There are no covalent bonds formed between the additive and the polymer. This is also demonstrated in Figure 6, which indicates there is no increase in molar mass (also see next response).

4

Is there an influence of the additive on the molar mass: please discuss

As stated above, the addition of BHET does not increase the molar mass of PLA, confirming the absence of covalent interactions. Indeed, Figure 6 shows a very small reduction in molar mass, which appears to be due to the scission of a small proportion of PLA chains. This proportion is, however, negligible and we consider it unlikely that there is any impact on crystallization behavior. These aspects are already discussed in the manuscript (see lines 234-236).

Round 2

Reviewer 1 Report

I reject to review this manuscript. As mentioned in previous review report, this manuscript should be reject.

Reviewer 3 Report

no further comments